# Immunization with *Anaplasma centrale* Msp2 HVRs Is Less Effective than the Live *A. centrale* Vaccine against Anaplasmosis

**DOI:** 10.3390/vaccines11101544

**Published:** 2023-09-29

**Authors:** Azeza Falghoush, Pei-Shin Ku, Kelly A. Brayton

**Affiliations:** 1Department of Veterinary Microbiology and Pathology, Washington State University, Pullman, WA 99164, USA; afalghoush@wsu.edu (A.F.);; 2College of Sciences, Sirte University, Sirte P.O. Box 674, Libya; 3College of Medical Technology, Aljufra University, Hun P.O. Box 61602, Libya

**Keywords:** bovine anaplasmosis, *A. centrale*, vaccine, immunization

## Abstract

Bovine anaplasmosis, caused by *Anaplasma marginale*, is the most prevalent tick-transmitted pathogen of livestock globally. In many parts of the world, *Anaplasma centrale*, a related organism, is used as a live blood-borne vaccine as it causes either no or only a mild clinical disease. *Anaplasma centrale* does not prevent infection with *A. marginale* but does prevent acute disease. *Anaplasma centrale* is prohibited from being used in the U.S. due to the risk of transmitting emerging pathogens. Both of these organisms encode proteins known as major surface protein 2 (Msp2), which is the most immunodominant protein for the organism. Both organisms persist in their host by evading clearance, i.e., the adaptive immune response, by recombining the hypervariable region (HVR) of *msp2* with pseudogene alleles. The study goal was to test whether the Msp2 HVRs encoded by *A. centrale* are a sufficient source of immune stimulation to provide the clinical protection exhibited by the blood-borne vaccine. Calves were inoculated with recombinantly expressed *A. centrale* HVRs. Control groups were inoculated with saponin or infected with the *A. centrale* live vaccine and compared with the test group. A Western blot analysis demonstrated that the HVR immunizations and *A. centrale* live vaccine stimulated an immune response. All animals in the study became infected upon challenge with *A. marginale*-infected ticks. The saponin-immunized control group had a high PPE (5.4%) and larger drops in PCVs (14.6%). As expected, the *A. centrale*-immunized animals were protected from acute disease with lower (0.6%) parasitemia and lower drops in PCV (8.6%). The HVR-immunized group had intermediate results that were not statistically significantly different from either the negative or positive controls. This suggests that the HVR immunogen does not fully recapitulate the protective capacity of the live vaccine.

## 1. Introduction

*Anaplasma marginale* is an intra-erythrocytic rickettsial hemoparasite that is transmitted by over 20 different species of ticks, including *Dermacentor andersoni*, and causes bovine anaplasmosis, a globally distributed disease that is of great economic impact to the cattle industry [1]. Clinical symptoms can include fever, anemia, abortions, weight loss, decreased milk production, and even death in some cases. Animals that survive acute disease remain persistently infected and serve as reservoirs for ongoing tick infection [2]. The pathogen evades the developing host immune response by means of two surface proteins, Msp2 and Msp3. Msp2 is the most immunodominant *A. marginale* protein, eliciting a strong primary and anamnestic response [2,3]. The single full-length *msp2* gene contains a central hypervariable region (HVR) that recombines with pseudogenes/donor alleles using gene conversion to generate new expressed variants that are not templated in the genome [4,5]. Over the course of infection, Msp2 has been observed to evolve from simple to complex variants that are mosaics of oligonucleotide segments from two or more pseudogenes that have recombined into the expression site [6]. These complex variants are thought to provide continued immune escape and sustain lifelong persistence in the host.

Control methods for anaplasmosis include treatment with tetracycline-based antibiotics, which have been ineffective at clearing infections [7,8]; and dipping with acaricides, which can lead to resistance in ticks and environmental contamination [9,10,11]. In the laboratory, outer membrane protein preparations have shown to be protective. However, these require technical expertise, are costly to produce, and are not viable for large-scale production [12,13,14]. Recently, a recombinant vaccine has been reported, but this requires further development prior to deployment [15]. Vaccines for anaplasmosis were reviewed in [16]. The most successful vaccine has been in use for over a century: a live blood-based vaccine using *Anaplasma centrale*, a close relative of *A. marginale*, which is used in many parts of the world to reduce severe symptoms and protect animals from death [17,18]. This vaccine has not been licensed in the United States due to the risk of spreading emerging agents [19]. Importantly, *A. centrale* has an *msp2* gene family that works similarly to the that of *A. marginale*: there is a single expression site and eight pseudogene/donor alleles that are used to generate variation and immune escape. The repertoire of pseudogenes is similar but distinct from those found in *A. marginale*. Due to the segmental gene conversion mechanism, the overall Msp2 HVR sequence identity is not a good measure of similarity; for example, within strain and between strain identities range from ~35 to 100%, and in the absence of an identical sequence, the identities top out at ~75% (see [20] Appendix A for full details). When comparing Msp2 HVR sequences between *A. marginale* and *A. centrale*, we see a range of similarity scores ranging between 31 and 72%, which is seemingly not that different when comparing only within *A. marginale*; however, for a key region in the HVR that is known as the Block I region, there is only one pseudogene in the *A. centrale* repertoire that contains the same sequence as any of the six known sequences found in this region for *A. marginale* [5,20,21,22,23]. In an effort to move towards a subunit vaccine, this study sought to test whether the protective capacity of the live *A. centrale* vaccine was primarily due to the Msp2 HVRs.

## 2. Materials and Methods

### 2.1. Msp2-HVR Vaccine Constructs

All unique pseudogene HVRs from the fully sequenced Israel strain of *A. centrale,* AcA1, AcF, AcB1, AcAF, AcG2, AcC, and AcA22 (AcG1 is identical to AcG2; Genbank accession no. CP001759) [21] were cloned into the entry vector pDONR221 and then into pDEST17, an expression vector with a N-terminal 6XHis tag (Invitrogen, Carlsbad, CA, USA). The primers are shown in Table 1. The clones were grown in LB media supplemented with 100 μg/mL carbenicillin; before reaching 0.4 OD_600,_ the cultures were induced with 1 mM IPTG. Cells were lysed with buffer containing 8 M urea, 50 mM sodium phosphate, and 0.3 M sodium chloride, pH 7.4, and the cells were washed with 8 M urea, 50 mM sodium phosphate, 0.3 M sodium chloride, and 10 mM imidazole, pH 7.4. Recombinant proteins were then purified by using HisPur^TM^ Cobalt Resin (Thermo Scientific, Waltham, MA, USA). The proteins were eluted with8 M urea, 50 mM sodium phosphate, 0.3 M sodium chloride, and 150 mM imidazole, pH 7.4. Purified recombinant proteins were loaded into a Slide-A-Lyzer dialysis cassette with a 3000 molecular weight cut-off (MWCO) (Thermo Scientific, Waltham, MA, USA) and dialyzed against phosphate-buffered saline (PBS). The eluted proteins were then dialyzed and concentrated by using Amicon Ultra filter units with a 3000 MWCO (Millipore, Burlington, MA, USA). Finally, the concentrated proteins were purified by size exclusion on a Superdex 75 column XK16 using an ÄKTA FPLC system (GE Healthcare, Chicago, IL, USA).

### 2.2. Animal Immunization

Nine Holstein calves were confirmed to be negative for *Anaplasma* antibodies using c-ELISA (VMRD, Pullman, WA, USA) prior to experimental infection [24]. The first group (animal nos. C42238, C42301, C42302) was immunized three times in 21-day intervals with a cocktail containing 20 ug of Msp2-HVR suspended in 2.5 mg of saponin at a total volume of 1 mL administered subcutaneously. The second group (animal nos. C42239, C42262, and C42299) was immunized with the *A. centrale* vaccine by intravenous inoculation with 2 mL of blood stabilate (1 mL of *A. centrale* stabilate mixed with 1 mL of FBS). The negative control group (animal nos.# C42236, C42324, C42331) received three injections of saponin adjuvant subcutaneously every 21 days.

### 2.3. Immunoblotting

To determine whether the immunizations were stimulating an immune response, Western blots were conducted, which were timed after the third immunization (for the HVR group) and prior to the challenge with *A. marginale* St. Maries (StM). Proteins were separated by SDS polyacrylamide gel electrophoresis using 4–20% precast gels (BioRad, Hercules, CA, USA) and were transferred to polyvinylidene fluoride (PVDF) membranes, which were placed in blocking buffer and shaken for 1 h. Bovine serum was collected from each animal and then diluted to 1:200. A total of 100 µL of antibody was added, and the membrane was shaken for an additional 1 h. The membranes were washed according to the established protocol and incubated with a 1:20,000 dilution of anti-bovine Ig2:HRP for 1 h with shaking, and the membranes were detected using the Amersham ECL Prime Western Blotting System (GE Healthcare, Chicago, IL, USA).

### 2.4. Infection of D. andersoni Ticks and Challenge

*Dermacentor andersoni* from Reynolds Creek stock were fed on an *A. marginale* StM-infected calf (C1401b1) to obtain infected ticks for the challenge. Ticks were acquisition-fed for 7 days and then held at 26 °C for 7 days to allow for digestion of the blood meal. Twenty ticks were allowed to attach and feed on both the immunized and control group cattle for an additional 7 days to transmit *A. marginale* StM. The challenge strain, *A. marginale* StM, was chosen because it is known to be tick-transmissible, virulent, and has had its genome completely sequenced (Genbank accession no. CP000030). The challenge was performed 21 days after receiving the third injection of the desired treatment.

### 2.5. Animal Monitoring

Following immunization and challenge with *A. marginale* StM, blood samples were collected daily from each animal throughout peak bacteremia. The level of bacteremia is expressed as the percent of parasitized erythrocytes (PPE). The levels of anemia were measured by capillary tube centrifugation to determine decreases in the packed cell volume (PCV) from baseline, which is approximately 35%. All protocols of the animal experiments were approved by the Institutional Animal Care and Use Committee at Washington State University, USA, in accordance with institutional guidelines based on the U.S. National Institutes of Health (NIH) Guide for the Care and Use of Laboratory Animals (ASAF 3386).

### 2.6. DNA Extraction, Cloning, and Sequencing of msp2

The Genomic DNA of *A. marginale* was extracted from bovine blood samples using the DNeasy Blood and Tissue kit from Qiagen (Germantown, MD, USA). Amplification was then conducted using forward 5′-TGG AGG AGC AAG GGT TGA AGT and reverse 5′-TTA CCA CCG ATA CCA GCA CAA primers to amplify the HVR region of the *msp2* gene for cloning and sequencing. PCR products were cloned in the PCR4-Topo vector (Invitrogen, Carlsbad, CA, USA). At least 30 colonies were selected from each cloning experiment (from each host animal) and grown overnight in LB broth for use in plasmid extraction. The GeneJET plasmid miniprep kit from Thermo Scientific was used to extract plasmid DNA. Vector NTI (Invitrogen, Carlsbad, CA, USA) and Snapgene (Mission City, CA, USA; available at snapgene.com) software were used to analyze the sequence data (Eurofins Genomics, Bothell, WA, USA) that were generated from the cloning experiment of each animal. Variants that differed by a single amino acid were counted as the same. All novel HVR variant sequences have been deposited in Genbank under accession numbers OR340832–OR340842.

### 2.7. Statistical Analysis

Data were analyzed using the Prism 8.0 software package (GraphPad, San Diego, CA, USA). Unpaired Student’s *t-*tests were used to test for statistical significance among immunized groups and the control. All statistical analyses were considered significant at *p* < 0.05.

## 3. Results

### 3.1. Antibody Response

Holstein calves were immunized with a cocktail of seven recombinant *A. centrale* Msp2 HVRs. As a positive control, calves were immunized with the live, blood-borne *A. centrale* vaccine while, as a negative control, calves were immunized with the adjuvant alone. All calves (three/group) were seronegative prior to immunization. Both *A. centrale*- and Msp2-HVR-immunized groups generated antibodies that recognized the recombinant proteins after immunization. A Western blot analysis showed that animals immunized with the cocktail of Msp2-HVRs appeared to have a higher titer of antibodies for the seven recombinant proteins and recognized all of the immunizing proteins (Figure 1). It is not surprising that the *A. centrale*-immunized animals did not detect all of the HVRs because the live vaccine may not have expressed all HVR variants for sufficient enough periods to induce a robust immune response. The adjuvant-immunized animals elicited no response to the recombinant proteins.

### 3.2. Clinical Response after Challenge

To challenge the cattle, 20 *A. marginale* exposed ticks were allowed to feed on each of the immunized animals for 7 days. All nine animals, regardless of vaccination group, became infected with *A. marginale* (Figure 2A). Infection parameters were assessed and compared between groups. There was not a significant difference between the days to peak parasitemia between groups. As expected, the negative control group animals had the highest bacteremia, with a mean (±SD) maximum peak percentage of infected erythrocytes of 5.4% (±1.25), while the *A. centrale* vaccinated group had the lowest bacteremia, with a mean (±SD) maximum peak percentage of infected erythrocytes of 0.52% (±0.003), which was significantly different from the negative control group (*p* = 0.003). Interestingly, the HVR-immunized animals had an intermediate level of bacteremia (PPE = 3.51% ± 2.37), which was not significantly different from either control group, reflecting incomplete protection with this vaccination regime.

Similarly, when examining the nadir of the PCV, the negative control group had the biggest drop in PCV, the *A. centrale*-immunized group had the smallest drop in PCV, and the HVR-immunized animals showed an intermediate result (Figure 2B). The difference between the negative control group and the *A. centrale* group was significantly different (*p* = 0.03), while the HVR group was not significantly different from either of the other two groups.

This intermediate result is because one of the HVR-immunized animals was protected in a manner that was indistinguishable from the *A. centrale*-immunized animals, while the other two acted similarly to the adjuvant control group (Figure 3).

### 3.3. Sequence Analysis of HVR Regions

Because one of the HVR-immunized animals (C42302) behaved differently from the other two upon challenge, we sought to determine if this was due to a different pattern of *A. marginale* Msp2 usage during the challenge. We examined which Msp2 alleles were expressed by amplifying, cloning, and sequencing expression site variants at peak parasitemia, rationalizing that these variants are the ones being expressed before an immune response is mounted to control the first wave of infection. A deduced protein alignment of the HVR sequences shows that there are 11 unique HVR variants represented in the study (Figure 4 and Table 2). The variants were scored for complexity [6], with complex variants typically being seen as a result of immune selection. Only four complex variants were identified, with an overall prevalence of 9% (Table 2) in the dataset. Most of the variants that were detected were simple variants; that is, the HVR was composed of two or fewer distinct pseudogene segments. Simple variant 1 was the most common variant and was detected in all animals. Variant 1 had a prevalence of 45% in the saponin control group, while in the *A. centrale-* and HVR-immunized groups, it was detected as 27% and 29% of the variants, respectively.

Although the overall prevalence of V1 in the unprotected animals is statistically significant compared with the protected animals, the numbers of this variant were identical between an HVR-immunized animal that was protected (C42302) and one that was not (C 42238).

One statistically significant difference that appears to be associated with protection is higher levels of expressed variant E6/F7 (corresponding to the HVR of pseudogene E6/F7 having recombined into the expression site). This variant was detected in 23–57% of the variants detected in the *A. centrale*-immunized animals and in animal C42302 (HVR-immunized, protected), while it only made up 1.4% of the detected variants in animals that were not protected in the saponin control group or the other two HVR-immunized animals (Table 2).

Variant 1, which was prevalent in all the animals after challenge, had identity scores ranging from 28 to 52% with the *A. centrale* Msp2 HVRs that were used for immunization, while the other variants had identity scores ranging from 30 to 72% (Appendix A). In other words, it appeared to be more diverse compared with the *A. centrale* HVRs than the other variants that appeared in the challenge.

Variant E6/F7 was found at higher levels in protected animals. This variant had an identity range of 40–57% with the *A. centrale* Msp2 HVRs, which was slightly more similar to the *A. centrale* Msp2 HVRs than the other variants detected at challenge (See Appendix A).

## 4. Discussion

We reject the hypothesis that the protective capacity of the *A. centrale* live vaccine is primarily due to immunity that is engendered by Msp2. While all of the *A. centrale* HVR vaccinates developed a strong humoral response, two of the three animals responded in a manner that was indistinguishable from the adjuvant only control, i.e., they were not protected. In the *A. centrale* live vaccine group, although all of the animals became infected, they were protected from acute disease.

We next examined whether the different treatment groups (i.e., *A. centrale* Msp2-HVR and *A. centrale* live vaccine) skewed the expression of Msp2 upon challenge with *A. marginale*. Interestingly, the most common variant detected was V1, which had the lowest similarity of any of the variants that were expressed during challenge when comparing across the HVR when compared with the *A. centrale* HVRs (Appendix A). This variant has been detected in previous *A. marginale* St. Maries strain infections and has been examined to show that it retains similar porin functionality as Msp2 molecules that are derived from full-length pseudogene HVRs [25]. The pseudogenes that are retained in the genome are subject to selection, whereas the transient recombined expression site mosaics that arise during the course of infection are not; therefore, it is notable that variant V1 is known to retain functionality.

While V1 was statistically more prevalent in the unprotected animals, this is likely due to the fact that the E6/F7 variant was able to establish in the protected animals. Overall, the E6/F7 variant is more similar to *A. centrale* Msp2 sequences than V1; however, it has a novel Block 1 sequence, which is likely a key factor in allowing organisms that express E6/F7 to be established.

However, robust protection as elicited in the animals that received the *A. centrale* live vaccine is likely to be multifactorial and continually boosted by the presence of the organisms that present more than just Msp2. It is known that both outer membrane protein (OMP) preparations and OMP complexes (a subset of the OMP preparations) from *A. marginale* are protective and can even elicit sterile immunity [12,13,26]. When immunizing with the OMP preparations or complexes, some of the proteins are naturally linked due to their context in the membrane. It has been demonstrated that linkage can be important in inducing immune responses; it was shown that a T-cell epitope in one protein enhanced a B-cell response in another linked protein [27]. In the study by Noh et al. [26], the authors used *A. marginale* OMP complexes that were covalently linked or unlinked to immunize cattle. Upon challenge, both groups were protected in a manner that was indistinguishable from each other. One difference between the two groups was that the linked immunogen group developed higher IgG2 titers prior to challenge. As protection was equivalent, this indicates that IgG2 is not a robust correlate of protection for *A. marginale*, as has been reported [14,28]. While Msp2 is the most immunodominant molecule on the surface of the organism, the subdominant epitopes of other surface proteins are also detected by the immune response [13,26]. In the search for a subunit vaccine, none of the recombinant proteins found in the OMP complexes that have been tested as vaccine candidates have appeared to be promising immunogens [29,30]. Vaccine formulation and or presentation may be just as important as having the right immunogen.

The challenge of working with the immunodominant Msp2 is that it has the ability to undergo gene conversion and evade the developing immune response. With the availability of genome sequence data, we have tracked infections and *msp2* usage [6] and we see that typically, the whole pseudogene HVRs are used early in infection. Therefore, the rationale to immunize with this set of HVRs and develop immunity against them and most variants that would arise in early infection was tested. Despite using HVR sequences that would provide the immunostimulatory capacity provided by *A. centrale* Msp2 early in infection, the Msp2-HVR vaccine did not provide protection that was equivalent to the live *A. centrale* vaccine. It may be that epitopes from the conserved region of Msp2 are required or that, as with other candidates that have been presented in recombinant form (mentioned above), the presentation is not appropriate to induce a protective response. Proteins presented in their native form appear to perform better in immunization trials, i.e., the OMP preparations and others [31,32,33].

This study makes it clear that Msp2 alone is insufficient for a robust protective response and that multivalent vaccines must be considered when moving away from the live *A. centrale* vaccine.

## Figures and Tables

**Figure 1 vaccines-11-01544-f001:**
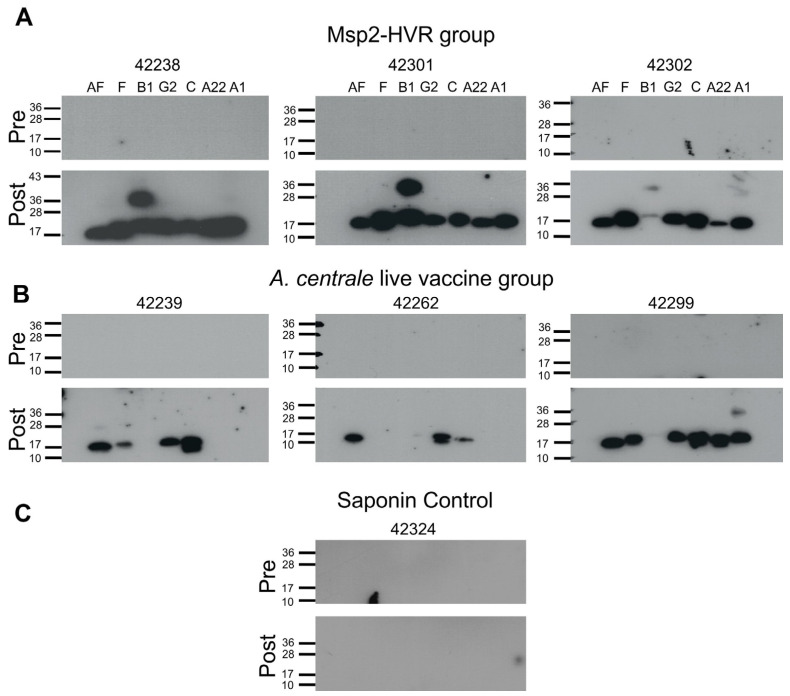
Western blots demonstrating antibody responses in vaccinates. Animals that were immunized with a cocktail of seven recombinant *A. centrale* Msp2 HVRs (**A**) or with live, blood-borne *A. centrale* (**B**) had an antibody response that recognized the recombinant proteins after immunization. There was no response to these proteins in the negative control group that was sham immunized with adjuvant (similar results from all animals; only the results from calf no. 42324 are shown) (**C**). All animals had no response prior to immunization (Pre).

**Figure 2 vaccines-11-01544-f002:**
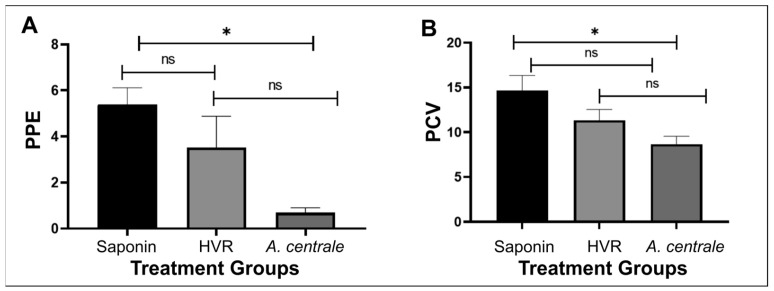
Clinical parameters for animals challenged with *A. marginale* StM. (**A**) The adjuvant-immunized group developed a statistically significantly higher infection based on the percent of parasitized erythrocytes (PPE) compared with the *A. centrale*-immunized group. (**B**) The *A. centrale*-immunized group had the smallest drop in the packed cell volume (PCV), which was statistically different from the adjuvant immunized group (*, *p* < 0.05; ns, not significant). Error bars represent the standard errors of the means (SEM) calculated from three animals in each group.

**Figure 3 vaccines-11-01544-f003:**
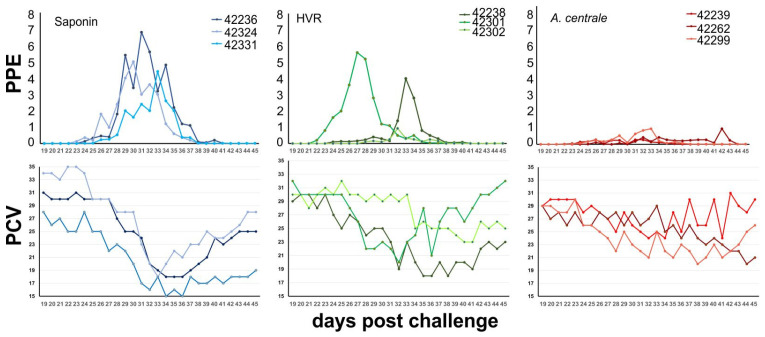
Clinical parameters in the challenged animals over time. Top graphs show recorded values of the percent of parasitized erythrocytes (PPE) for each animal plotted by days post challenge. The protected animal from the HVR-vaccinated group (# 42302) had similar values of daily counted infected erythrocytes to that of the *A. centrale*-vaccinated animal group (red). The adjuvant-immunized group (blue) and the two other unprotected HVR-immunized animals developed the highest PPE. The lower graphs show the packed cell volume for each animal over time. The animals are color-coded the same in both sets of graphs.

**Figure 4 vaccines-11-01544-f004:**
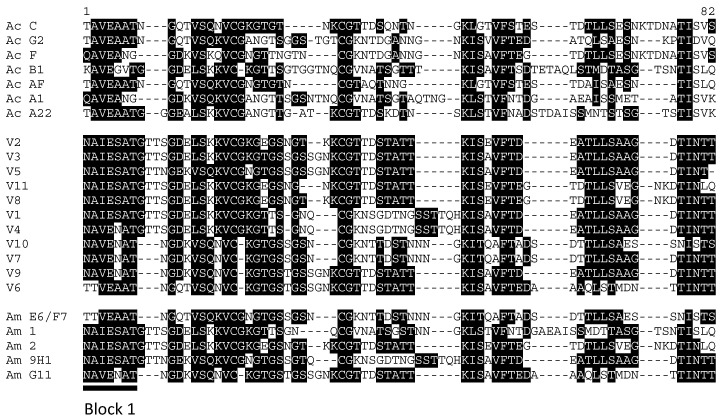
Alignment of Msp2 HVR sequences. Unique HVR sequences from *A. centrale* (Ac) are shown in the top group. The middle group are the expression site variants detected in the study that are not templated in the genome. The final group is composed of sequences that represent the unique pseudogene HVRs found in the *A. marginale* St. Maries strain genome, all of which were expressed and detected during the challenge phase of the study.

**Table 1 vaccines-11-01544-t001:** Primers used to clone *A. centrale* pseudogene HVRs.

Locus-Specific Primers		
ψ name	Primer Name	Primer Sequence *
AcA1 AcF	A1, F-Fx *attB*2	GGGGACAAGTTTGTACAAAAAAGCAGGCTATCAGGCAGTAGAGGCTAAT
AcA1	A1-Rx *attB*2	GGGGACCACTTTGTACAAGAAAGCTGGGTACTTCACGCTGATGGTGGC
AcB1	B1-Fx *attB*1	GGGGACAAGTTTGTACAAAAAAGCAGGCTATAAAGCAGTAGAAGGTGTT
AcB1 AcAF	B1, AF-Rx *attB*2	GGGGACCACTTTGTACAAGAAAGCTGGGTACTGCAAGCTGATGGTGTT
AcAF AcG2 AcC	AF, G2, C-Fx *attB*1	GGGGACAAGTTTGTACAAAAAAGCAGGCTATACTGCAGTAGAGGCTGCC
AcG2	G2-Rx *attB*2	GGGGACCACTTTGTACAAGAAAGCTGGGTACTGCACGTCGATGGTGGG
AcA22	A22-Fx *attB*1	GGGGACAAGTTTGTACAAAAAAGCAGGCTATACTGCAGTAGAGGCTGCT
	A22-Rx *attB*2	GGGGACCACTTTGTACAAGAAAGCTGGGTACTTCACGCTGATGGTGCT
AcF AcC	F, C-Rx *attB*2	GGGGACCACTTTGTACAAGAAAGCTGGGTATGACACACTGATGGTAGC

* *msp2*-specific sequences are underlined.

**Table 2 vaccines-11-01544-t002:** Sequence analysis of Msp2 usage and HVR unique variants detected during the challenge.

Number Clones from Each Animal				
Saponin	HVR	*A. centrale*	Variant Name	Type of Variant
20, 22, 17 *^,a^	14, 12, 12 *^,b^	13, 7, 15 *^,c^	V1	Simple
0, 0, 1	1, 1, 0	0, 0, 1	V2	Complex
6, 1, 0	1, 2, 3	0, 0, 1	V3	Complex
0, 3, 0	ND ^#^	1, 0, 1	V4	Complex
1, 1, 0	ND	ND	V5	Simple
0, 0, 2	ND	ND	V6	Simple
0, 0, 1	ND	ND	V7	Complex
ND	1, 0, 0	ND	V8	Simple
ND	0, 4, 0	ND	V9	Simple
ND	ND	1, 0, 0	V10	Simple
ND	ND	0, 0, 1	V11	Simple
1, 0, 2	4, 2, 0	2, 0, 1	ψ 1	*A. marginale* StM
2, 1, 3	1, 0, 0	3, 0, 2	ψ 2	*A. marginale* StM
0, 2, 2	8, 8, 2	2, 0, 1	ψ G 11	*A. marginale* StM
0, 0, 1	0, 1, 9	8, 17, 7	ψ E6 F7	*A. marginale* StM
0, 0, 1	0, 0, 4	0, 1, 0	ψ 9H 1	*A. marginale* StM
ND	ND	0, 5, 0	ψ A22	*A. centrale*

* Calf numbers (in order): ^a^ Saponin group C# 42324, C# 42236, C#42331; ^b^ *A. centrale* Msp2 HVR-immunized group C# 42238, C# 42301, C# 42302; ^c^ *A. centrale*-immunized group C# 42239, C# 42262, C# 42299; ^#^ ND = not detected.

## Data Availability

The data presented in this study are available in the body of this article as well as Appendix A. Sequence data has been deposited in Genbank under accession numbers OR340832–OR340842.

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
