# Peer review of "Immunization with Anaplasma centrale Msp2 HVRs Is Less Effective than the Live A. centrale Vaccine against Anaplasmosis"

_vaccines, 2023, doi:10.3390/vaccines11101544_

Round 1

Reviewer 1 Report

The authors have tested the protective capacity of recombinant MSP2 hypervariable regions of A. centrale against A. marginale challenge in cattle. The study has a clear objective and it is executed well with attention to detail. The authors have observed strong protection in animals vaccinated with live A. centrale vaccine and protection close to live vaccine one animal from MSP2 hypervariable region (HVR) vaccinated group. The authors have further looked at the variation in HVR’s in study animals to understand the differential protection in HVR vaccinated group. The manuscript is acceptable for publication after minor revision. I hope the authors would continue their future work to understand the differential protection in HVR vaccinated group further.

Lines 88-90: Have the authors used mix of male and female calves or one sex?. It will be interesting to see whether any sex related factors involved in the protection observed in one animal in HVR vaccinated group.

Line 107: Please mention the secondary antibody dilution instead of antibody volume.

Line 155: Can the authors please mention the time point of sera used in the western blot?  

Line 156-157: The western blots images shown in Supplementary data appears to show some reactivity at higher molecular weight than HVR’s and just wanted to check whether it is background reaction with E. coli proteins or some non-specific reaction? Please add a comment about this in the results if the authors think it is appropriate. Also, please add molecular weight marker details in the supplementary figure as well.

Lines 169-174: Can the authors please provide the order of HVR’s in the western blot either in the figure or in the figure legends? Also, the authors have provided clear images in the supplementary data and the image provided in the manuscript seems to be not very clear and appears like it has water mark around the bands. Can the authors please provide clear images in the main manuscript as well?

Lines 195-197: It is great to see one animal (42302) performing very similar to live vaccine. The western blot image appears to show the protected animal has less antibodies than non-protected animals and especially against 3 and 6th antigens loaded in the gel. Have the authors looked at the antibody levels against different HVR’s in the vaccinated animals using an ELISA?

Author Response

We thank the reviewer for their comments. 

The authors have tested the protective capacity of recombinant MSP2 hypervariable regions of A. centrale against A. marginale challenge in cattle. The study has a clear objective and it is executed well with attention to detail. The authors have observed strong protection in animals vaccinated with live A. centrale vaccine and protection close to live vaccine one animal from MSP2 hypervariable region (HVR) vaccinated group. The authors have further looked at the variation in HVR’s in study animals to understand the differential protection in HVR vaccinated group. The manuscript is acceptable for publication after minor revision. I hope the authors would continue their future work to understand the differential protection in HVR vaccinated group further.

Lines 88-90: Have the authors used mix of male and female calves or one sex?. It will be interesting to see whether any sex related factors involved in the protection observed in one animal in HVR vaccinated group.

We typically use only male animals in our experiments because the female animals are used in the dairy, while males are excess animals. Furthermore, this was a preliminary experiment, with small group sizes, and with 3/group, it would have been difficult to sufficient numbers of each sex to be able to draw conclusions based on sex.

Line 107: Please mention the secondary antibody dilution instead of antibody volume.

Corrected

Line 155: Can the authors please mention the time point of sera used in the western blot?  

Sera was taken after the 3rd immunization for immunoblots, this is stated in the methods; but as all animals did not receive 3 immunizations it is simply stated as “western blots were conducted prior to the challenge with A. marginale St. Maries (StM)”

Line 156-157: The western blots images shown in Supplementary data appears to show some reactivity at higher molecular weight than HVR’s and just wanted to check whether it is background reaction with E. coli proteins or some non-specific reaction? Please add a comment about this in the results if the authors think it is appropriate. Also, please add molecular weight marker details in the supplementary figure as well.

  • As required by the journal, the original western blots were provided to show that we had not manipulated them. They are not meant to be included as supplemental material. 
  • The smudges I believe the reviewer is referring to are indeed background/non-specific reactions as evidenced by the fact that they are present in the blots with the preimmunization serum.

Lines 169-174: Can the authors please provide the order of HVR’s in the western blot either in the figure or in the figure legends? Also, the authors have provided clear images in the supplementary data and the image provided in the manuscript seems to be not very clear and appears like it has water mark around the bands. Can the authors please provide clear images in the main manuscript as well?

I have updated Figure 1 with the loading order of HVRs.

I am happy to try to provide better images, but the images as I see them look as intended, so I am not sure what the reviewer is seeing.  Can the editorial staff please advise.  I have also provided tiff images in the original submission.

Lines 195-197: It is great to see one animal (42302) performing very similar to live vaccine. The western blot image appears to show the protected animal has less antibodies than non-protected animals and especially against 3 and 6th antigens loaded in the gel. Have the authors looked at the antibody levels against different HVR’s in the vaccinated animals using an ELISA?

No, we did not do ELISAs.  Standardization would be difficult, and the potential for cross reaction makes interpretation difficult.

Reviewer 2 Report

The postulated hypothesis in this manuscript proposes that antibodies to A. centrale’s MSP2 hypervariable regions are capable of protecting against challenge with live A. marginale. 

The first assumption states that A. centrale MSP2 hipervariable regions (HVR) are similar to those of A. marginale and that immunity to these HVR protect against A. marginale. It is true that these HVR are similar (not identical). Results show that authors were right, immunity induced by inoculation with recombinant proteins representing the A. centrale HVR shared between the two organisms do induce production of antibodies that cross-react between the two species. Immunization with MSP2 HVR recombinant proteins was first explored by Noh et al., (Vaccine 28 (2010) 3741–3747). These authors immunized animals with St Maries recombinant MSP2 conserved regions (CR) and HVR and challenged vaccinees with St. Maries live agent. Their results show that while the levels of antibodies were similar, only antibodies to CR correlated with protection. The fact that inoculation with rHVR proteins induce production of antibodies is not strange as MSP2 has been described as highly immunogenic and perhaps immunodominat protein. MSP2 has been attributed functions in evasion of the immune response, inducing variants, which arise in a cyclic manner every 6 to 8 weeks. The use of the variants as immunogens is an idea that has merit, but as observed in other studies as time passes, immunity to previous MSP2 variants do not necessarily protect against the new ones. In the present work, animals were immunized with A. centrale MSP rHVR proteins representing a number of antigenic variants and antibodies produced against these variants do not seem to protect against live A. marginale challenge.

The second assumption –as understood by the present reviewer- is that A. centrale immunity against A. marginale would be based solely or mainly on MSP2 HVR. Vaccine studies published over many years have shown that the use of one or a few proteins may induce immunity against homologous/autologous challenge, yet immunity against a heterologous challenge is harder to attain except when the proteins were inserted in cross-linked A. marginale membranes (e.g. Macmillan et al., 2008). As shown by Noh et al., (2010) protection at least against A. marginale MSP2 correlated more with Conserved Regions proteins than with HVR’s. Thus, not unexpectedly, immunizing with MSP2 variants induces incomplete protection at best.

The immunity generated this way also lacks the perspective of the cellular immune response as no T-cell epitopes were included in the immunogen.

It is clear that the authors pretended to induce an immune response, which may afford protection not against infection but at least against clinical signs. In this line, authors state that calves were used in these experiments. The authors do not describe accurately the age of the experimental subjects, thus, it is reasonable to assume that the subjects were in the range below 9 months of age. It is a well-known fact that calves under 9 months of age are usually refractive to developing clinical sings of the disease (Salinas et al., 2022). The values of clinical parameters of vaccinated (both live and HVR) and unvaccinated subjects’ are consistent with this premise. The values of highest rickettsemia and lower packed cell volume would not endanger the lives of any experimental subject. Furthermore, values within groups are disparate, particularly in the HVR vaccinees. The authors do not mention any other clinical signs in HVR- or non-vaccinated animals or if subjects received specific chemotherapy to alleviate clinical signs.

The authors measured antibody titers, yet, make no mention of the isotype produced. Brown et al, (1998) stablished that a Type 1 cellular immune response where IgG2 is prevalent (in contrast to IgG1), in protected animals against challenge. This finding was confirmed by Barigye et al., (2004) showing that animals which responded with a Type 1 cellular immune response were fully protected in contrast to animals which developed a Type 2 cellular immune response which required treatment to avoid death.

The present work as it is, is not different from previous reports, so its contribution should be seen from a different perspective, i.e. not if the immunity against A. centrale MSP2 HVR’s is capable of protecting against A. marginale, but rather if the immune response induced by A. centrale MSP2 HVR’s coincide with A. marginale.

In order to complete the picture of the immunity developed by the vaccinees the authors can also try to verify the type of antibodies (IgG1 vs IgG2) produced against live A. centrale HVR at vaccination. This view can shed some light into why only one animal was immune at challenge.

Author Response

We thank the reviewer for their comments. 

The postulated hypothesis in this manuscript proposes that antibodies to A. centrale’s MSP2 hypervariable regions are capable of protecting against challenge with live A. marginale. 

The first assumption states that A. centrale MSP2 hipervariable regions (HVR) are similar to those of A. marginale and that immunity to these HVR protect against A. marginale. It is true that these HVR are similar (not identical). Results show that authors were right, immunity induced by inoculation with recombinant proteins representing the A. centrale HVR shared between the two organisms do induce production of antibodies that cross-react between the two species. Immunization with MSP2 HVR recombinant proteins was first explored by Noh et al., (Vaccine 28 (2010) 3741–3747). These authors immunized animals with St Maries recombinant MSP2 conserved regions (CR) and HVR and challenged vaccinees with St. Maries live agent. Their results show that while the levels of antibodies were similar, only antibodies to CR correlated with protection. The fact that inoculation with rHVR proteins induce production of antibodies is not strange as MSP2 has been described as highly immunogenic and perhaps immunodominat protein. MSP2 has been attributed functions in evasion of the immune response, inducing variants, which arise in a cyclic manner every 6 to 8 weeks. The use of the variants as immunogens is an idea that has merit, but as observed in other studies as time passes, immunity to previous MSP2 variants do not necessarily protect against the new ones. In the present work, animals were immunized with A. centrale MSP rHVR proteins representing a number of antigenic variants and antibodies produced against these variants do not seem to protect against live A. marginale challenge.

The second assumption –as understood by the present reviewer- is that A. centrale immunity against A. marginale would be based solely or mainly on MSP2 HVR. Vaccine studies published over many years have shown that the use of one or a few proteins may induce immunity against homologous/autologous challenge, yet immunity against a heterologous challenge is harder to attain except when the proteins were inserted in cross-linked A. marginale membranes (e.g. Macmillan et al., 2008). As shown by Noh et al., (2010) protection at least against A. marginale MSP2 correlated more with Conserved Regions proteins than with HVR’s. Thus, not unexpectedly, immunizing with MSP2 variants induces incomplete protection at best.

The reviewer has misunderstood what was done in the paper by Noh et al.  In that paper they immunized with outer membrane protein preps (OMP preps) or outer membrane protein complexes (fewer proteins than OMP preps).  They examined responses to Msp2 as gauged against peptides to the CR or HVR.  However, given the complex immunogen, many factors are going into what induced protection in these animals, and thus, this experiment is not comparable.  While in this study, protection may have correlated with response to the CR peptides, other factors are also at play. 

The study by Macmillan simply shows the importance of T cell epitopes – and that they do not necessarily need to be in a given protein to help induce responses to that protein if the two proteins are naturally linked, such as Msp1a and Msp1b – but this is not particularly relevant to the present paper.

The immunity generated this way also lacks the perspective of the cellular immune response as no T-cell epitopes were included in the immunogen.

It is clear that the authors pretended to induce an immune response, which may afford protection not against infection but at least against clinical signs. In this line, authors state that calves were used in these experiments. The authors do not describe accurately the age of the experimental subjects, thus, it is reasonable to assume that the subjects were in the range below 9 months of age. It is a well-known fact that calves under 9 months of age are usually refractive to developing clinical sings of the disease (Salinas et al., 2022). The values of clinical parameters of vaccinated (both live and HVR) and unvaccinated subjects’ are consistent with this premise. The values of highest rickettsemia and lower packed cell volume would not endanger the lives of any experimental subject. Furthermore, values within groups are disparate, particularly in the HVR vaccinees. The authors do not mention any other clinical signs in HVR- or non-vaccinated animals or if subjects received specific chemotherapy to alleviate clinical signs.

In almost all the work we do we use calves, under 9 months of age, for a few reasons: 1) it is not cost effective to rear animals for a year to be able to do experiments, and 2) this experimental model has proved to be useful in many publications from my institution over many years (see the collective works of Guy Palmer and Wendy Brown et al.) and clinical signs are reported and 3) the goal is not to endanger the lives of the animals, but to see a clinically reportable difference, which is possible using calves. 

The reason we did not mention chemotherapy to alleviate clinical signs is because we didn’t use any, and we did not report what we did not do.

The authors measured antibody titers, yet, make no mention of the isotype produced. Brown et al, (1998) stablished that a Type 1 cellular immune response where IgG2 is prevalent (in contrast to IgG1), in protected animals against challenge. This finding was confirmed by Barigye et al., (2004) showing that animals which responded with a Type 1 cellular immune response were fully protected in contrast to animals which developed a Type 2 cellular immune response which required treatment to avoid death.

The present work as it is, is not different from previous reports, so its contribution should be seen from a different perspective, i.e. not if the immunity against A. centrale MSP2 HVR’s is capable of protecting against A. marginale, but rather if the immune response induced by A. centrale MSP2 HVR’s coincide with A. marginale.

I do not understand this comment.

In order to complete the picture of the immunity developed by the vaccinees the authors can also try to verify the type of antibodies (IgG1 vs IgG2) produced against live A. centrale HVR at vaccination. This view can shed some light into why only one animal was immune at challenge.

Actually IgG2 is not a good correlate of protection. While the reviewer is correct about the above cited papers, there are others that indicate that IgG2 production to Msp2 does not correlate with protection (ie Noh et al., 2010)

Reviewer 3 Report

The manuscript is well written, here are some suggestions. 

1) The title could be more specific?: Efficacy of Anaplasma centrale Msp2 hypervariable regions compared to the live A. centrale vaccine demonstrates the latter to be more protective. - Or something like that

2) line 12 - there is a 'full stop' after Anaplasma.

3) line 23 - (instead of 'upon tick challenge') - upon challenge with Anaplasma marginale positive ticks. (as currently A. marginale is not mentioned in the abstract at all)

4) The reference style is mixed - please check whole document for correctness in style and referencing - some references are not cited or in the wrong order.

a) line 35 - it is in brackets and not superscript.

b) line 51 has [3-5] and 16 in superscript.

c) after 10-12 is cited, the next number is '16', then '19' follows '17' etc..

d) line 283 has '142413' - that is a lot of references!

Author Response

We thank the reviewer for their comments.

We have changed the title of the manuscript to:

"Immunization with Anaplasma centrale Msp2 HVRs is less effective than the live A. centrale vaccine against anaplasmosis"

We have made all grammatical corrections as suggested.

We have corrected the references (with a lot of difficulty! ;-)).

Round 2

Reviewer 2 Report

The revised title states that immunization with Anaplasma centrale Msp2 HVRs is less effective at inducing protective immunity compared to the live A. centrale vaccine against anaplasmosis. However, the title lacks clarity regarding the aspect of reduced effectiveness, specifically whether it pertains to the vaccine's ability to confer protection or its capacity to stimulate an immune response. The inference from the results suggests that the reduced effectiveness is primarily related to its ability to induce an immune response that inhibits the development of bacteremia. Perhaps the title should be written in a manner that it reflects this fact.

This inference is drawn from the fact that infecting calves below 9 months of age with live A. marginale typically results in their survival, even in the absence of prior immunization.

The authors have commendably included detailed clinical data for all animals post-exposure to live A. marginale. However, a noticeable gap exists in their report as it lacks descriptions of the clinical parameters of the animals during the A. centrale infection/vaccination phase. While this omission might seem inconsequential at first glance, it becomes pertinent due to the close temporal proximity of the A. centrale infection and the subsequent A. marginale challenge. Such data would be instrumental in comparing the effects of vaccination versus the challenge and understanding the holistic impact of these interventions on the animals' health and immune responses.

It is worth considering whether presenting the actual PCV values before and after the A. marginale challenge might offer a more informative perspective. Instead of representing the results solely as a percentage drop in Packed Cell Volume, displaying the real PCV values before and after the A. marginale challenge could provide a more meaningful insight into the condition of the experimental subjects.

Figure 2A, the y axis reads PEE, and the figure legend reads Percent Parasitized Erythrocytes thus this axis should read PPE instead (see y axis in figure 3).

Figure 3 illustrates the fluctuations in PPE (Percent Parasitized Erythrocytes) over the 40-day challenge period. Notably, the authors have not addressed whether, during the assessment of bacteremia resulting from the A. marginale challenge in A. centrale vaccinated animals, any A. centrale-infected erythrocytes were influencing or contributing to the observed PPE values. This omission is significant as it could impact the accuracy of the interpretation of the PPE data and the overall assessment of protection conferred by the A. centrale vaccination against A. marginale challenge. Clarifying this point would enhance the rigor and completeness of the study's findings.

Lines 191 to 196: Building upon my previous comment, it's worth noting that even in adult animals, PPE values of 8% or lower and a 15% drop in PCV (Packed Cell Volume) might not typically manifest as noticeable clinical symptoms or acute disease. To illustrate this further, based on the information provided in the Materials and Methods section, let's assume an initial normal PCV value of 35%. A 15% PCV loss, as observed in the saponin group, relative to the initial value, would result in a final actual PCV of 29%. Importantly, this final PCV value still falls within the normal range, as indicated by Paré et al. in their 1993 study (Can J Vet Res 57:241–246). Therefore, Figure 2 may present a perspective that amplifies differences in PCV values, which in a real-world scenario could be even less significant or potentially go unnoticed clinically.

The rest of the manuscript based on the expression of different HVR variants and how would they relate to induction of immunity.

The conclusions of the manuscript are consistent with the results, i.e. immunization with A.c. HVR are not only not better at protecting, but they induce variable antibody responses at least in the three animals used in this group. The contention that animals were protected is based on two parameters, PPE and PCV both of which at the levels detected in immunized and control animals are not consistent with acute disease as described in discussion.

Thus, the conclusion should be that the immune response developed in HVR-immunized animals did not inhibit development of bacteremia rather that protect from acute disease, which, was not observed even in control animals.

Consequently, the primary conclusion that can be drawn is that the immune response elicited in HVR-immunized animals did not effectively inhibit the development of bacteremia. This underscores the importance of considering the broader context of the disease and its clinical manifestations when evaluating the effectiveness of immunization strategies.

Author Response

Response:

We changed the title as requested by reviewer 3.  We feel that the title is a fair reflection of the work within.  If the academic editor feels a different title is warranted, we are happy to entertain another change.

Reviewer 2 appears to be focused on the age of the calves, however, calves the world over become infected with A. marginale.  In a study we did in South Africa, they were infected in the first month of life.  This has little to do with what this study is about, we can still discern differences in responses.  If the animals die, there is nothing to study, so to have a study with death as the endpoint defeats the purpose.

The important feature of the A. centrale vaccination group is that they were protected.  That data is shown.  The saponin control shows that the challenge was good.

We have corrected Figure 2, Y axis to read PPE.  Thanks for catching this.

We have added a graph of the PCVs to Figure 3.  We separated out the 3 groups to do this to more easily see the groups.

Thanks for the comments.